# Shaping Sequence Attractor Schema in Recurrent Neural Networks

**Zhikun Chu[1]**
chukunzhi@outlook.com

**Bo Hong[3]**
hongbo@tsinghua.edu.cn

**Xiaolong Zou[3],✉**
benzou@mail.bnu.edu.cn

**Yuanyuan Mi[2],✉**
miyuanyuan@tsinghua.edu.cn

1, Bioengineering College,School of Medicine, Chongqing University.
2, Department of Psychological and Cognitive Sciences, Tsinghua University, Beijing, China.
3, Biomedical Engineering,School of Medicine, Tsinghua University.
✉: Corresponding authors.

## Abstract

Sequence schemas are abstract, reusable knowledge structures that facilitate rapid adaptation and generalization in novel sequential tasks. In both animals and humans, shaping is an efficient way to acquire such schemas, particularly in complex sequential tasks. As a form of curriculum learning, shaping works by progressively advancing from simple subtasks to integrated full sequences, and ultimately enabling generalization across different task variations. Despite the importance of schemas in cognition and shaping in schema acquisition, the underlying neural dynamics at play remain poorly understood. To explore this, we train recurrent neural networks on an odor-sequence task using a shaping protocol inspired by well-established paradigms in experimental neuroscience. Our model provides the first systematic reproduction of key features of schema learning observed in the orbitofrontal cortex, including rapid adaptation to novel tasks, structured neural representation geometry, and progressive dimensionality compression during learning. Crucially, analysis of the trained RNN reveals that the learned schema is implemented through sequence attractors. These attractor dynamics emerge gradually through the shaping process: starting with isolated discrete attractors in simple tasks, evolving into linked sequences, and eventually abstracting into generalizable attractors that capture shared task structure. Moreover, applying our method to a keyword spotting task shows that shaping facilitates the rapid development of sequence attractor schemas, leading to enhanced learning efficiency. In summary, our work elucidates a novel attractor-based mechanism underlying schema representation and its evolution via shaping, offering new insights into the acquisition of abstract knowledge across biological and artificial intelligence.

## 1 Introduction

Imagine taking the subway in a new station: you can effortlessly anticipate the sequence of events - entering the station, purchasing a ticket, scanning it, waiting for the train, and boarding. This ability stems from an abstract knowledge structure encoded in your brain [1], which organizes the typical order and relationship among these events, commonly referred to as a schema. Schemas facilitate rapid learning [2, 3], flexible decision-making [4], and efficient generalization [5, 6, 7], forming the foundation of cognitive flexibility and generalization in both animal and human intelligence [3]. How-

39th Conference on Neural Information Processing Systems (NeurIPS 2025).

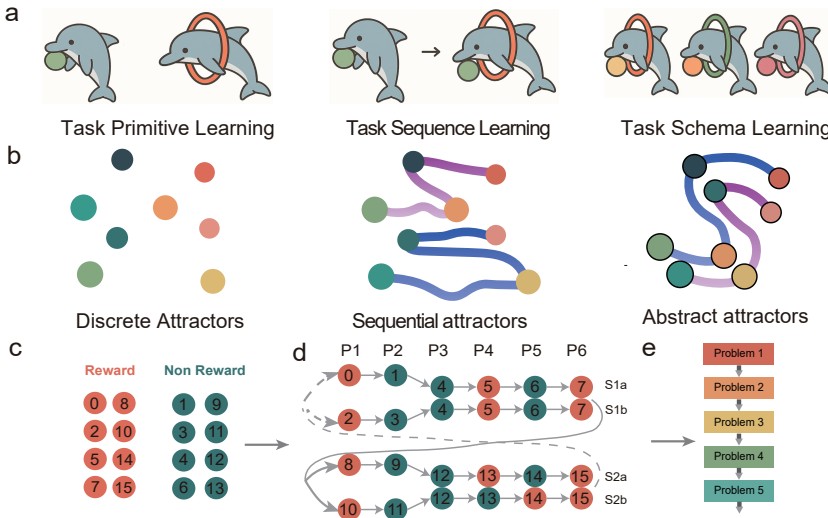

Figure 1: Schematic of learning sequence schema via Shaping. (a) A dolphin example. Task primitive learning: the dolphin learns basic actions independently, such as holding a ball and passing through a hoop. Task sequence learning: it learns to order these actions temporally - holding the ball first, then passing through the hoop. Task schema learning: exposure to different ball and hoop types allows the dolphin to develop generalizable schemas, enabling adaptation to similar tasks. (b) A neural dynamical hypothesis. Task primitive learning: Neural systems acquire basic attractor structures (e.g., discrete attractors). Task sequence learning: These discrete attractors are temporally linked into structured sequence attractors. Task schema learning: These sequence-specific attractors are compressed into an abstract schema. (c) Task primitive learning: An odor-reward association task is presented where 16 odors are linked to different rewards (red = rewarded, blue = non-rewarded). (d) Task sequence learning: Using the 16 odors, an odor-sequence task is constructed with two pairs of six-position sequences (S1a-S1b and S2a-S2b). Arrows show transitions between and within sequences. (e) Task schema learning: Five new problems with the same sequence structure but novel sets of 16 odors are introduced consecutively.

ever, despite their importance, the neural mechanisms underlying the representation and acquisition of schemas remain poorly understood.

For simple tasks, schemas can be learned through direct trial-and-error training on related experiences. However, for more complex sequential tasks, this approach often fails. Instead, animals and humans typically rely on shaping - a process that decomposes complex tasks into simpler subtasks learned incrementally [8, 9, 10]. As illustrated in Fig. 1a, teaching a dolphin to jump through a hoop while holding a ball exemplifies this approach. The animal first learns each basic skill independently (e.g., jumping through a hoop, holding a ball), then practices performing them in sequence, and ultimately generalizes the behavior across task variations (e.g., different ball colors or hoop types). Schema learning via shaping typically unfolds in three stages [4]: (1) task primitive learning, where basic action components are acquired; (2) task sequence learning, where these components are integrated into structured behavioral sequences; and (3) task schema learning, where abstract regularities are extracted across multiple structurally related tasks. While shaping has proven effective in practice, how it supports the learning and representation of task schemas remains unclear.

Schemas, primarily represented in the prefrontal cortex (PFC) [1, 3], are characterized by two key properties: low dimensionality and compositionality. The PFC is thought to support rich, low-dimensional attractor dynamics that underlie flexible behavior - for example, continuous attractor dynamics for sensory integration [11] and discrete attractors for sensorimotor transformations [12]. These low-dimensional dynamics have been proposed as neural representations of schemas [13] and are thought to underlie the learning-to-learn phenomena observed in primates [5]. Schemas are also compositional, allowing them to be reused and recombined to solve novel tasks [1]. For instance, rats can transfer learned spatial schemas from one context to novel memory tasks [2]. Moreover, recent

computational studies demonstrate that basic attractor dynamics, arising from multitask learning, can serve as primitive schemas that flexibly combine to facilitate new task learning [29, 15].

Building on these propoties, we propose a dynamical perspective on schema formation through shaping in complex sequential tasks (Fig. 1b), consisting of three stages: (1) Task primitive learning establishes basic attractors for individual task components, such as discrete attractors; (2) Task sequence learning integrates them into structured sequential dynamics, such as sequential attractors; (3) Task schema learning abstracts and compresses these patterns into unified low-dimensional representations that capture shared temporal and structural regularities.

Although the dynamic view of schema formation is compelling, it lacks a concrete computational model and direct comparison with neural data. In this work, we use recurrent neural networks (RNNs) to investigate how schema representations emerge and evolve through shaping. We validate our approach in an odor-sequence task widely used to study schema learning in the orbitofrontal cortex (OFC) [4]. Our main results and contributions are as follows.

(1) Our shaping-trained RNN systematically replicates key features of schema learning observed in rats' OFC ensembles, including faster learning on novel tasks, structured task representation geometry, and progressive dimensionality compression.

(2) The model reveals that schema representations emerge as low-dimensional sequence attractors, composed of discrete attractors linked by some low-dimensional manifolds, offering a mechanistic account of schema encoding and testable predictions for neuroscience.

(3) Dynamic analysis uncovers how attractor structures evolve during shaping: from isolated discrete attractors to integrated sequence attractors, to abstract, compact attractor structure - providing a novel dynamical mechanism of schema formation.

(4) Extending our sequence attractor-based shaping approach to the keyword spotting task shows improved learning efficiency, demonstrating its potential in complex, real-world tasks.

Together, our work advances the understanding of schema representation and its learning via shaping from a dynamical systems perspective, offering new insights into how abstract knowledge arises from experience through curriculum-like processes in both neuroscience and machine learning.

## 2 Experimental Background

To model schema learning under shaping conditions, we employ the odor-sequence task of schema evolution from Zhou et al. [4], a study designed to investigate this process in rats using a typical shaping protocol. In each task instance (termed a "problem"), 16 novel odors are randomly sampled and organized into two pairs of sequences (S1a – S1b and S2a – S2b), shown in Fig. 1d. In each of the six positions within a sequence, a single odor is presented, and the rat must decide if it signals a reward. Transitions among the sequences S1a, S1b, S2a, and S2b are selected at random. Although odor identities change in each new problem, the core sequence and reward structures remain the same. Thus, rats need to learn this invariant task structure for rapid adaptation. Zhou et al. adopted a shaping paradigm to train rats to learn the challenging task. (1) Task primitive learning: Initial training on a basic odor-reward association task with 16 randomly sampled odors (Fig. 1c). (2) Task sequence learning: Introduction of a structured sequence task using the same 16 odors and the predefined template (Fig. 1d), requiring the rat to link learned associations into temporal sequences. (3) Task schema learning: Introduction of five new problems successively, each with a novel set of 16 odors following the same task structure (Fig. 1e). The main findings of Zhou et al. are twofold: First, rats exhibit increasingly faster learning of new problems with more experience. Second, population activity in the OFC becomes progressively lower-dimensional and forms increasingly structured patterns that reflect the underlying task schema. However, the precise neural representation of schemas for odor-sequence tasks and the dynamics of their evolution through shaping are still unclear.

## 3 Methods

### 3.1 Model Structure

We employ an RNN model to demonstrate schema learning via shaping on the odor-sequence task. The RNN comprises three components: an input layer with $M$ units, a recurrent layer with $N$ units,

and an output layer with $K$ units, as illustrated in Fig. 2a. At time $t$, given an input stimulus vector $I_m(t)$, the total input $x_n(t)$ to recurrent unit $n$ evolves according to the following dynamics:

$$\tau \frac{dx_n(t)}{dt} = -x_n(t) + \sum_{j=1}^{N} W_{nj}^{\text{rec}} r_j(t) + \sum_{m=1}^{M} W_{nm}^{\text{in}} I_m(t), \tag{1}$$

here, $W^{\text{in}}$ and $W^{\text{rec}}$ represent the input-to-recurrent and recurrent-to-recurrent weight matrices, respectively. The activation of the unit $n$ is computed as $r_n(t) = \tanh(x_n(t))$, and $\tau$ denotes the time constant.

The output $y_k(t)$ of unit $k$ is given by:

$$y_k(t) = \sum_{n=1}^{N} W_{kn}^{\text{out}} r_n(t). \tag{2}$$

The output layer linearly reads out the RNN's activity to generate task-relevant predictions, including odor classification and reward prediction, depending on the shaping stage. The recurrent-to-output weight is denoted by $W^{out}$. The RNN is simulated using Euler's method of numerical integration.

## 3.2 Task Design and Shaping Procedure

In the odor sequence task, each odor stimulus is represented as a one-hot vector of dimension $M = 96$, with the non-zero entry set to 20 (see Appendix Sec.A for details). Each stimulus presentation lasts $T_s = 5$ steps. Following the shaping paradigm of Zhou et al. [4], the RNNs undergo training across three stages, detailed below.

**Task Primitive Learning**. In this stage, RNNs are trained on a simple prediction task that learn to associate individual odors with corresponding reward outcomes. Each 15-step trial includes a 5-step delay, a 5-step odor stimulus, and a 5-step delay, and is corrupted by Gaussian white noise with a mean of 0 and a variance of 1. The OFC flexibly encodes task-relevant variables, including odor rewards and identities [16, 17]. As both are required in this task, the model's output layer comprises $K = 18$ units: a 2-dimensional reward prediction vector $\hat{y}^{\text{reward}}(t)$ and a 16-dimensional odor classification vector $\hat{y}^{\text{class}}(t)$. For rewarded odors, $\hat{y}^{\text{reward}}(t) = [2, 0]$; for non-rewarded odors, $\hat{y}^{\text{reward}}(t) = [0, 2]$. Odor identity is encoded as a one-hot vector in $\hat{y}^{\text{class}}(t)$. Output units are required to be at zero before the odor stimulus appears, transition to target values during presentation, and remain there until it ends (Fig.S1). This target sequence mimics the ramping dynamics observed in cortical circuits during decision-making [18]. Finally, the loss function used during task primitive learning is defined as a regression loss:

$$L_1 = \frac{1}{2} \sum_{t=1}^{T} \left\| y_{0:2}(t) - \hat{y}^{\text{reward}}(t) \right\|^2 + \beta \sum_{t=1}^{T} \left\| y_{2:K}(t) - \hat{y}^{\text{class}}(t) \right\|^2. \tag{3}$$

The total loss is balanced between reward prediction and odor identity classification by a weighting factor $\beta$, which we set to 0.5 here.

**Task Sequence Learning**. During this stage, RNNs, having been pretrained in the task primitive learning phase, undergo further training in an odor-sequence problem. As described in Sec. 2 and Fig. 1d, the same 16 odors from the previous stage are organized into four sequences. Taking S1a as an example, each 90-step sequence trial comprises six odor stimuli presented at intervals sampled from a uniform distribution between 5 and 10 steps, mimicking experimental variability [4]. To simulate sensory noise, the entire input stream is corrupted with Gaussian noise (mean = 0, variance = 1). As odor identities in the sequence task become unreliable predictors of reward (illustrated by odor 13 indicating different rewards at P4 and P5), the network shifts its prediction target to only reward outcomes. Consequently, the output layer is simplified to $K = 2$ units, representing only the reward prediction vector $\hat{y}^{\text{reward}}(t)$, with the same values as in the task primitive learning setup. At the start of the sequence, all output targets are set to zero. Upon stimulus presentation, the outputs transition to their target values and hold them until the next odor appears, at which point they update accordingly. The loss function is defined as:

$$L_2 = \frac{1}{2} \sum_{t=1}^{T} \left\| y(t) - \hat{y}^{\text{reward}}(t) \right\|^2. \tag{4}$$

**Task Schema Learning**. As shown in Fig. 1e, the pretrained RNN from the previous stage is further trained on a series of odor-sequence problems sharing the same underlying structure. For each task, 16 new one-hot vectors with the active entry set to 20, different from those in the previous task, are selected from the 96-dimensional odor space. All other task parameters, including sequence duration and background noise level, remain consistent with those in task sequence learning. The network continues to predict only reward outcomes, with the loss defined as:

$$L_3 = \frac{1}{2} \sum_{t=1}^{T} \left\| y(t) - \hat{y}^{\text{reward}}(t) \right\|^2. \tag{5}$$

Across all three stages, we use the Adam optimizer. See Appendix Sec.A for more training details.

## 4 Results

### 4.1 Learning-to-learn and Structured Task Representation Emerge in Shaping-Trained RNNs

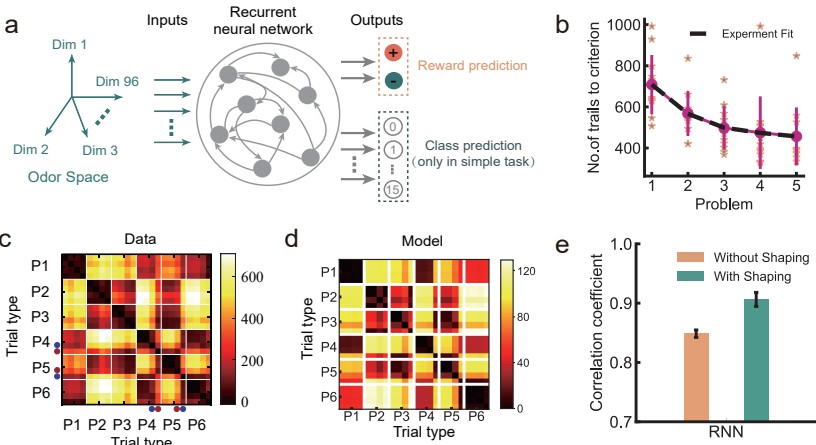

Figure 2: Behavior and representation in RNNs via shaping. (a) RNN architecture. A three-layer network (input, recurrent, output). The input is a 96-dimensional odor stimulus vector. The output layer predicts reward and odor class. (b) During task schema learning, the number of training trials for the shaped RNN to reach a performance criterion across consecutively presented problems (10 seeds per problem).The dashed line represents the polynomial fit. (c-d) Dissimilarity matrices. Representational dissimilarity matrices (RDM) of OFC ensembles(c) [4] and RNN hidden activities (d). Each matrix shows the distance between the neural/model representations of all pairs of trial types (24 total, 6 per sequence) across sequences and positions. (e) Pearson correlation between the experimental and model RDMs. Blue indicates RNNs trained with shaping, and pink indicates those without. Results are averaged over 10 seeds. See Appendix Sec.A for detailed parameters.

Schemas are proposed to encode abstract, generalizable knowledge, enabling faster learning in novel situations [2, 4]. To demonstrate this, we train an RNN using the shaping process described in Sec. 3.2. The RNN first learns a simple odor-reward association task until the training loss falls below 0.05, followed by an odor-sequence task using the same odors, again trained until the loss falls below 0.05. It is then exposed to a series of novel tasks that share the same task structure but involve entirely new stimuli. Importantly, no explicit meta-learning objective is used. During this task schema learning phase, the RNN exhibits learning-to-learn behavior: the number of trials required to reach the criterion decreases significantly across problems (Fig.2b). An exponential fit reveals a clear trend of accelerated learning, suggesting the RNN utilizes prior knowledge for more efficient adaptation. This pattern mirrors learning-to-learn behavior observed in rats performing the odor-sequence task.

A hallmark of schema learning observed in rats' OFC is the emergence of structured neural representations [4], as shown in Fig.2c. To assess this in our model, we replicate the experimental analysis by computing pairwise distances between trial types in the RNN's hidden state space across positions and sequences, yielding an $24 \times 24$ RDM (see Appendix Sec.C.1 for details). This RDM reflects the

structured geometry of the task representation space. As shown in Fig.2c,d, the RNN's RDM closely aligns with that observed in the OFC. Quantitatively, the Pearson correlation between the model and the experimental RDM exceeds 90% (Fig.2e), indicating that the RNN captures the hierarchical task representation structure and exhibits a representational geometry similar to OFC (see Fig.S2). To test the role of shaping, we conduct an additional control experiment in which the RNN is trained directly on a set of odor-sequence problems without shaping. This control model exhibits markedly lower correlation with OFC ensembles than shaped RNNs, indicating that shaping is important for the emergence of structured task representation in the RNN.

## 4.2 Low-Dimensional Sequence Attractors Serve as Schema for Generalization

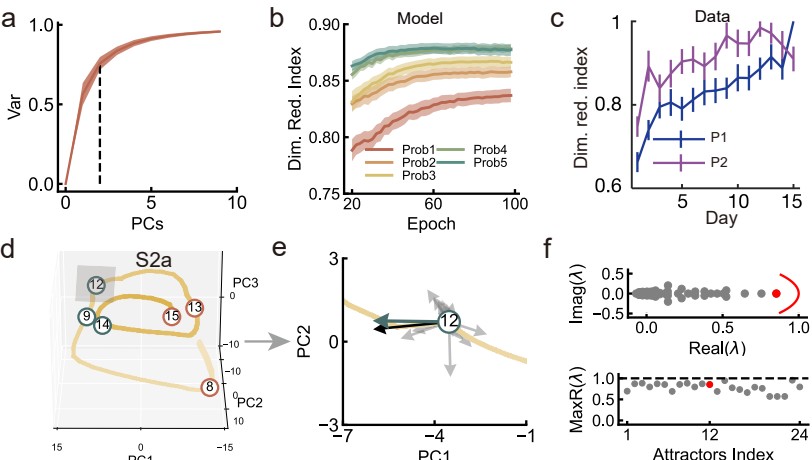

Figure 3: Dimensionality and Attractor Dynamics in a Shaped RNN. (a) Cumulative variance of RNN activity explained by principal components (PCs). Dashed line marks the top three PCs.(b) Dimensionality compression over time in the RNN across five odor-sequence problems. Dim.red.index, dimensionality reduction index - is the normalized variance explained by the top three PCs. Shaded areas indicate the standard error of the mean averaged across 20 seeds. (c) Dimensionality compression over time in rat OFC across two odor-sequence problems [4].(d) PCA visualization of neural trajectories for four sequences (S1a, S1b, S2a, S2b) in the shaped RNN, projected onto the top three principal components. Six fixed points per sequence were identified via optimization-based analysis. (e) Zoom-in of the shaded area in (d). Black arrow: empirical transition direction at the odor 12 fixed point. Green arrow: principal eigenvector; gray arrows: other eigenvectors of the Jacobian matrix at odor 12. (f) Top: Eigenvalue spectrum of the Jacobian at the odor 12 fixed point. Bottom: Maximum eigenvalues across all 24 fixed points, with the red dot indicating the one corresponding to odor 12. See Appendix Sec.A for detailed parameters.

Our brains can reduce the dimensionality of neural representations during schema formation by averaging unique problem details while preserving cross-problem commonalities [3]. To examine this in our RNN, we perform PCA on hidden activities during task schema learning. Fig. 3a shows the network dynamics reside in a low-dimensional subspace, with the top three PCs explaining nearly 80% of the variance. We further track dimensionality change over learning. Fig. 3b shows that the explained variance of the top 3 PCs progressively increases within and across problems, indicating dimensionality compression of neural activity over learning, similar to OFC activity in rats (Fig. 3c).

Together, these findings suggest that a low-dimensional schema code emerges in RNNs trained via shaping, evident in both behavior and representation. However, the underlying neural dynamics of this code remain unclear. To investigate this, we visualize the RNN's state dynamics using PCA. As shown in Fig.3d and Fig.S3, the four sequences (S1a, S1b, S2a, and S2b) form distinct low-dimensional trajectories in the RNN state space. Using optimization-based fixed point analysis [15], we identify six fixed points along each trajectory, each fixed point corresponding to a specific odor's position within the sequence (detail methods see Appendix Sec.C.2). Sequences sharing similar reward and transition structures (e.g., S1a, S1b, S2a) have close trajectories, indicating abstraction of representations with structural commonalities. To probe the stability of these fixed points, we

examine the eigenvalues of the local Jacobian matrix (Fig. 3f). All maximal eigenvalues are below 1, indicating that the fixed points are locally stable and act as attractors. Notably, the principal eigenvector - associated with the largest eigenvalue - aligns with the direction of sequential transitions (Fig. 3e), suggesting it mediates movement along the trajectory.

These results reveal that the RNN encodes odor sequences as sequence attractors - sequences of stable fixed points connected by low-dimensional manifolds. Through a perturbation study (Fig.S4), we further demonstrate that this sequence attractor schema serves as a reusable flexible template, enabling new odor cues to bind and thereby facilitating efficient generalization in novel situations.

## 4.3    Gradual Emergence of Sequence Attractor Schema Through Shaping

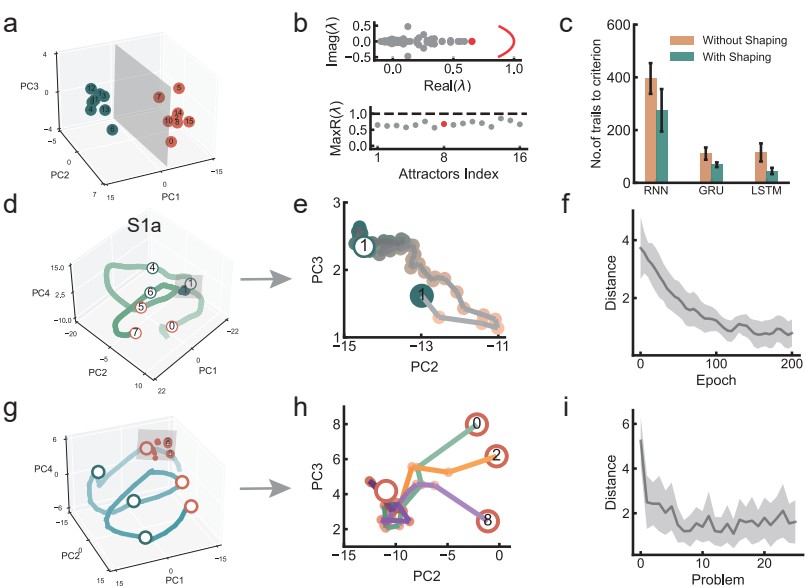

Figure 4: Evolution of attractor dynamics in RNNs during shaping. (a - c) Task primitive learning. (a) PCA visualization of identified fixed points; red = rewarded states, blue = non-rewarded. (b) Stability analysis of fixed points: top, eigenvalue spectrum of a sample odor 8; bottom, maximal eigenvalues across fixed points. (c) Learning efficiency in the subsequent sequence learning stage, comparing models with and without prior task primitive learning, averaged over 5 seeds. Note that the total trial number to criterion includes both task primitive and sequence learning stages for a fair comparison. (d - f) Task sequence learning. (d) PCA visualization of attractor dynamics reveals the gradual shift of the discrete attractor for odor 1 (primitive learning) toward its position in the sequence attractor associated with S1a. Green curve: neural trajectory of S1a. Solid circle: prior attractor from task primitive learning; hollow circles: attractors of S1a sequence. (e) Zoom-in on the learning trajectory of the odor 1 attractor. (f) The average Euclidean distance between evolving attractor states and their corresponding locations of sequence attractors (task sequence learning) over training epochs (averaged over 5 seeds). (g - i) Task schema learning. (g) PCA visualization of attractor abstraction. Green curve: converged abstract neural trajectory of S1a, S1b, and S2a. Numbered hollow circles: prior attractors (sequence learning). Unnumbered hollow circles: attractors of the abstract trajectory (task schema learning). (h) Zoom-in of the abstraction process in (g), illustrating the migration of prior attractors: odor 0 at S1a, odor 2 at S1b, and odor 8 at S2a (sequence learning), toward the abstract attractor within the final sequence attractors (schema learning). (i) The average Euclidean distance between attractors of corresponding sequence steps in S1a, S1b, and S2a, tracked during problem training (averaged over 5 seeds). See Appendix Sec.A for detailed parameters.

Given that sequence attractors serve as a schema, we next investigate their evolution during shaping. We analyze attractor dynamics across three shaping stages. First, during task primitive learning (Fig. 4a, b), the RNN learns simple odor-reward associations and develops 16 discrete, locally stable attractors, each representing a specific odor cue. Importantly, RNNs initialized with these prior discrete attractors learn the subsequent task sequence significantly faster, requiring fewer trials to

reach criterion (Fig. 4c). This learning benefit is also consistent across diverse recurrent architectures, such as LSTM and GRU.

During task sequence learning, we track how prior attractor states in the RNN's state space evolve and examine whether they're reused to form sequence attractors, as shown in Fig. 1b. At each training step, we probe the RNN's attractor states. Taking odor 1 as an example, we initialize the hidden state to its prior attractor and iterate the network dynamics without input until convergence to a new stable state. This new state is then used as the initial state for the next training batch iteration, repeating this process to generate the evolving trajectory of stable states. Fig. 4d,e illustrate the gradual migration of odor 1's prior attractor (primitive learning) to its corresponding location within the S1a sequence attractor structure (sequence learning). We found that 87% (14 out of 16) of prior attractors successfully evolve to their correct target locations. Quantitatively, the average distance between these successfully evolving states and their corresponding attractor locations (sequence learning) steadily decreases over training (Fig. 4f, detail methods see Appendix Sec.C.3). These findings suggest that attractors learned during task primitive learning are reused, reorganized, and linked to form sequence attractors during sequence learning.

During task schema learning, we examine how sequence-specific attractors are compressed into abstract sequence attractors. Figures 4g,h illustrate this process: the attractors for odor 0 in S1a, odor 2 in S1b, and odor 8 in S2a gradually converge toward a common attractor. The average distance between attractors occupying the same sequence position across S1a, S1b, and S2a also decreases over training (Fig. 4i). These results suggest that as the network learns related problems presented consecutively, it gradually averages out sequence-specific details while retaining commonalities shared across S1a, S1b and S2a - eventually forming a unified abstract sequence attractor. This convergence is expected, given the common reward transition structure across these sequences.

Together, these findings reveal a dynamic, attractor-based mechanism for schema formation through shaping. First, discrete attractors emerge during task primitive learning. These are then organized into sequence-specific structures during task sequence learning. Finally, in task schema learning, these structures are compressed and abstracted into a unified schema. This sequence attractor-based schema shaping process supports our conceptual idea shown in Fig. 1b.

## 4.4 Divergent Learning Dynamics in RNNs with and without Shaping

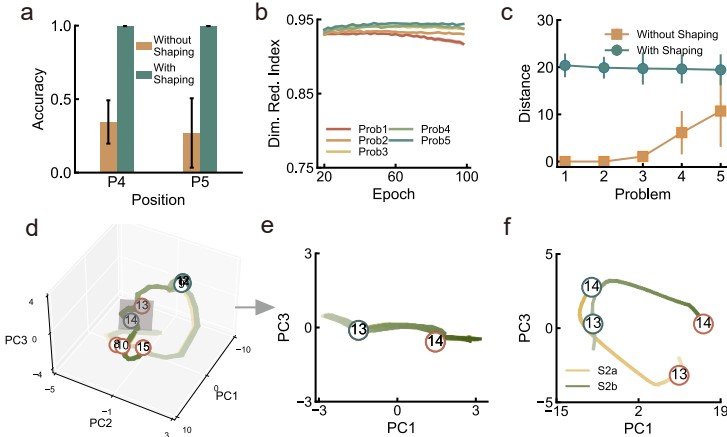

Figure 5: Evolution of attractor dynamics in RNNs without shaping. (a) Performance comparison of RNNs with and without shaping at P4 and P5 in S2a/S2b. Networks were trained and evaluated after problem 1 learning (averaged over 5 seeds). (b) Dimensionality compression dynamics in RNNs without shaping across five odor-sequence problems (averaged over 5 seeds). Other settings are identical to Fig. 3b. (c) Average Euclidean distance between the evolving attractor states corresponding to P4 and P5 across S2a and S2b sequences for RNNs with and without shaping during problem training (averaged over 5 seeds). (d) PCA visualization of sequence attractors in a representative RNN without shaping. Sequence attractors in S2a and S2b converge into a single trajectory after problem 1 learning. (e) Zoomed-in view of attractor states for P4 and P5 in S2a and S2b. (f) PCA visualization of attractor states at the same corresponding positions in problem 5 as shown in (e).

To elucidate the mechanisms underlying behavioral differences between shaped and unshaped RNNs, we examine the learning dynamics of unshaped networks. Shaped RNNs achieve markedly higher reward prediction accuracy ($99.89 \pm 0.03\%$) than unshaped ones ($93.20 \pm 2.4\%$), with the largest discrepancy in classifying odors at P4 and P5 of S2a/S2b, where reward contingencies are reversed. As shown in Fig. 5a, shaped RNNs reach near-perfect accuracy, whereas unshaped RNNs reach only 25% after problem 1. Applying the same PCA-based dimensionality analysis as in Fig. 3b (Fig. 5b), we find that the dimensionality reduction in unshaped RNNs differ qualitatively: (1) Premature compression – unshaped networks enter a low-dimensional regime (93%) much earlier than shaped ones (82.5%), indicating an early collapse to coarse representations; and (2) Lack of progressive abstraction – unlike shaped networks, unshaped ones show little change in dimensionality across problems. These findings reveal fundamentally distinct learning dynamics between shaped and unshaped RNNs.

We further analyze and visualize sequence attractors to better understand representational evolution. In shaped networks, the task-primitive learning stage already establishes four distinct attractors for the odor events at P4 and P5 (in S2a and S2b). These attractors serve as stable building blocks for higher-level, sequence-specific attractor trajectories in later stages, maintaining a consistent distance between corresponding attractor states (Fig. 5c). In contrast, unshaped networks lack this scaffold. Early in training, they represent P4 and P5 odors in S2a and S2b with overlapping trajectories, and solving the task therefore requires the later formation of separate attractor trajectories - a process that is often unstable and leads to failed convergence. Fig. 5d–f shows a successful example of late attractor trajectory separation, where the trajectory for problem 5 diverges from the shared trajectory established after learning problem 1.

In summary, shaped networks build schemas progressively using scaffolded attractor dynamics and structured compression, whereas unshaped networks exhibit abrupt, shallow compression that leads to poor generalization. These findings demonstrate that shaping fundamentally alters the learning trajectory and attractor organization of RNNs, clarifying how and why shaping facilitates learning.

## 4.5 Sequence Attractor-Based Shaping Improves Learning Efficiency in Keyword Spotting

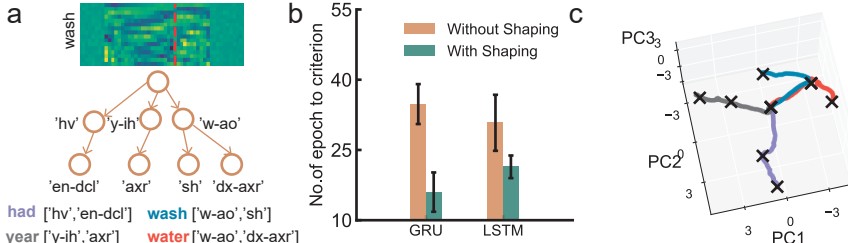

Figure 6: Sequence attractor-based shaping in a keyword spotting task. (a) The task involves four keywords: "water," "wash," "had," and "year." The upper panel shows the MFCC representation of a sample utterance of "water," while the lower panel illustrates the tree-structured phoneme organization of keywords. (b) Learning efficiency comparison (epochs to $90\%$ accuracy) between models trained with and without shaping, averaged over 5 seeds. (c) PCA visualization of RNN neural trajectories after shaping. Crosses indicate identified fixed points. See Appendix Sec.D.2 for detailed parameters.

Based on the sequence attractor-based shaping framework, we apply our method to a real-world sequence task: keyword spotting (Task details see Appendix Sec.D.1). The task involves four spoken words - "water," "wash," "year," and "had" - each composed of basic phonemes forming a hierarchical structure (Fig. 6a). These words are randomly selected from the TIMIT dataset [19]. Following the shaping approach, we divide the task into three stages. In task primitive learning, the model learns to classify phonemes using only one sample per word, forming discrete phoneme-level attractors. In task sequence learning, the model links these attractors by training on sampled word sequences for word class prediction. Finally, in task schema learning, the model trains on the full dataset to abstract a schema and generalize across word variations.

As shown in Fig.6b, shaping significantly improves learning efficiency compared to a model trained without shaping, requiring fewer epochs to reach $90\%$ test accuracy. PCA visualization of hidden

states reveals a tree-like structure consistent with the phoneme hierarchy, supporting the emergence of a sequence schema (Fig.6c). Additionally, we identify seven leaf attractors corresponding to individual phonemes, and one root attractor representing the initial state prior to phoneme onset. These results demonstrate that the sequence attractor-based shaping framework generalizes to complex sequence learning tasks, offering both interpretability and efficiency.

## 5 Discussion

Sequence schemas form the core of flexible and generalizable intelligence across animals and humans. For complex tasks, acquiring such schemas through pure trial-and-error is often ineffective. Instead, animals and humans use shaping - breaking down complex tasks into simpler subtasks and gradually assembling the full schema. However, the underlying dynamic mechanism behind this process remains unclear. In this work, we use RNNs to study how schemas are represented and evolve through shaping. Our main findings are fourfold: (1) We systematically replicate key behavioral and neural features of schema learning observed in the OFC of rats. (2) We show that sequence schemas can be encoded as sequence attractors. (3) We identify a novel dynamic process of schema formation, progressing from point attractors to sequence attractors, and finally to abstract schemas. (4) We demonstrate the practical utility of this framework by successfully applying it to keyword spotting. These findings may advance understanding of the neural mechanisms of schema learning via shaping and offer new insights into how abstract, generalizable structures can be learned from experience in both biological and artificial systems.

**Related Works**. Schema learning has attracted increasing attention in neuroscience [3, 10, 13, 15, 20, 21, 22, 23, 27]. Previous studies have primarily examined schema representations - for example, low-dimensional manifolds or attractors in sensorimotor [13], decision-making tasks [15], or capturing temporal order [20]. However, a fundamental question remains overlooked: how does the brain learn schemas through shaping? Despite its prevalence in animal training, the mechanism of shaping is systematically underexplored in neuroscience. And previous works mainly focus on behavioral outcomes, neglecting the underlying dynamics of abstract knowledge acquisition [21, 22]. Our work addresses this deficit by investigating schema learning via shaping, providing the first explicit link between these two essential concepts.

Our approach mirrors curriculum learning by decomposing complex tasks into simpler subtasks to boost learning efficiency [24, 25]. However, while curriculum learning in machine learning primarily focuses on optimizing performance generalization [25, 26], it typically lacks a mechanistic understanding of how the underlying abstract knowledge is acquired during the process [25]. In contrast, we offer a dynamic framework grounded in biological learning: learning progresses from primitive attractor structures to reusable abstract attractor schemas that facilitate generalization [15]. This framework may provide inspiration for new curriculum learning algorithms that explicitly facilitate the learning and use of abstract knowledge.

Recent studies have questioned the identifiability and robustness of RNNs in neuroscience modeling, showing they can achieve similar behaviors via distinct internal solutions [28]. Our analysis, however, reveals that shaped RNNs converge to highly similar internal dynamics regardless of initialization. Applying Canonical Correlation Analysis (CCA) to models trained with different seeds (Fig.S6), we find strong alignment among the top three canonical components, indicating consistent representation geometry across models. Previous findings show that task complexity or multi-task learning can reduce degeneracy in RNN solutions [28, 29], we extend this by demonstrating that our shaping paradigm may be an effective way to reduce this degeneracy.

**Limitations and Future Works**. A limitation of this work is that model optimization relies on backpropagation through time, which is not biologically plausible [30]. Future work could explore combining dopamine-based reward learning [31] with slow Hebbian plasticity [32] to develop more biologically grounded mechanisms for schema learning. Another limitation is that our model includes only the PFC, whereas schema formation in the brain is thought to result from dynamic interactions between the PFC and the hippocampus [17, 33, 34], which play complementary roles with distinct learning timescales [35]. The hippocampus may support task primitives and sequence learning, while the PFC contributes to task schema learning. Modeling their interaction could help clarify the neural basis of schema formation through shaping and inspire more flexible and generalizable sequence learning algorithms.

## Acknowledgments and Disclosure of Funding

This work was supported by the National Science and Technology Innovation 2030 Major Program (No. 2021ZD0203700 / 2021ZD0203705, Y.Y. Mi), National Natural Science Foundation of China (62336007, B. Hong).

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
