# OpenReview forum: "Shaping Sequence Attractor Schema in Recurrent Neural Networks"
_NeurIPS.cc/2025/Conference — NeurIPS 2025 poster_

### Official Review · Reviewer_nFhk · 2025-07-01

**Clarity:** 3
**Significance:** 2
**Originality:** 2
**Rating:** 4
**Confidence:** 4

**Summary:**

The authors present a method for training recurrent neural networks (RNNs) on an odor-reward association task using a ‘shaping procedure’. This procedure, inspired by how rats are trained on the same task, is implemented in three stages. The authors show that shaping leads to faster learning compared to training without shaping, promotes the development of structured task representations, and results in a progressive reduction in the dimensionality of network activity over training. They further analyze the network dynamics, showing a transition from isolated discrete attractors to sequential attractors, and eventually to more abstract sequence attractors. This progression is proposed as a novel perspective on schema formation in biological neural systems. The approach is also applied to a keyword spotting task, where similar patterns emerge, supporting the generality of the findings.

**Questions:**

(1) Figure 2(e) shows that shaping improves the correlation between the representational dissimilarity matrices (RDMs) of the model and the data. However, even without shaping, the model still achieves a relatively high correlation and also, a high correlation does not immediately mean that it is a well structured representation. Could you explore or provide another way to demonstrate that shaping is essential for developing a well-structured representation? For example, is there a qualitative difference in the types of attractors or representations formed with vs. without shaping?

(2) The paper shows that shaping leads to a reduction in the dimensionality of the network's activity. Could you also show whether a similar reduction does not occurs in models trained without shaping? This would clarify whether the reduction in dimensionality is specifically a result of shaping or a more general consequence of learning.

(3) In the first stage of training, the RNN is trained to output the odor identity in addition to reward/non-reward. This seems to differ from the animal experiment or were rats also required to classify all 16 odors, or was the task limited to a binary choice (rewarded vs.  non-rewarded)? If the animals were not trained to output odor identity, please explain more clearly why this added output was included in the model and why the comparison to experimental behavior remains valid. Also, would the shaping procedure still be effective if the model were only required to predict reward/non-reward, as in the behavioral task?

(4) You state that shaping leads to more efficient or faster learning (learning-to-learn, Fig., 2(b)). Did you control for how many entirely new odors are introduced in each subsequent problem? If the number of novel odors decreases over time, this could itself contribute to faster learning as the task effectively get easier (less new odors) rather then the network better at learning. Please consider adding a control to distinguish these effects.

(5) L215ff: In this section, you analyze attractor convergence by turning off all inputs. However, since the attractors depend on input, this approach may not reflect where the attractors are under task-relevant conditions. Please clarify this analysis and how it relates to behaviorally meaningful network states. Also, does this approach imply that you are training or tuning the initial conditions of the network during training? If so, how does this relate to the experimental setup with rats?

**Ethical Concerns:**

["NO or VERY MINOR ethics concerns only"]

**Final Justification:**

The authors provide a compelling computational account of shaping and schema formation, grounded in dynamical systems and aligned with biological data. Their rebuttal clarified conceptual and methodological points, added useful analyses, and strengthened links to existing work. The additional experiments address concerns around the necessity and impact of shaping, supporting the main claims. While some limitations remain, this is a valuable contribution.

**Limitations:**

(see questions above)

**Paper Formatting Concerns:**

no major formatting issues

**Quality:**

2

**Strengths And Weaknesses:**

The structuring of shaping as a learning path is interesting and well thought out. It's particularly nice that the approach is grounded in experimental data and aims to closely follow and be compared to the animal training protocol. However, in its current state it provides only limited evidence that shaping is crucial to achieve the observed properties such as abstract structured representations but mainly shows that also with shaping these properties emerge.

The concept of shaping described in the paper is closely related to curriculum learning in machine learning, where tasks are broken down and gradually made more difficult over training to facilitate and guide learning. This connection is currently not very clearly referenced or discussed.

The paper is well structured and some minor clarifications would help to avoid some misunderstandings:
* Lines 84 and following: It would be helpful to state more clearly that the initially learned values associated with different odors do not persist throughout the sequences. This point only becomes fully clear later in the text or by closely analyzing the figure, which cause a lot of confusion for me.
* Please clarify what specific behavior was expected from the rat at each training stage. A more explicit description would make it easier to compare the experimental protocol with your model and better understand how the shaping stages align.
* Line 124: You mention that Gaussian noise was added, but it would improve clarity to include this term directly in Equation 1 (and/or Equation 2), so the reader can see clearly how noise is incorporated into the model dynamics.

---

> ### Author Rebuttal · Authors · 2025-07-30
>
> We thank the reviewer for valuable comments .We'll address each of your concerns point by point.
>
> ***On Weaknesses:***
>
> 1. Thank you for the valuable feedback. Shaping plays a critical role in two key ways:
>
> (1) It accelerates learning, as shown in Fig. 4c, 5b, and S4.
> (2) It enables the emergence of abstract structured representations robustly that are difficult to reach via direct training, as detailed in reply in question 1 (see below).
>
> Specifically, in networks without shaping, we observe that odors at P4 and P5 in S2a and S2b often collapse onto shared attractor trajectories and leads the network to stack at a local minima. In contrast, shaping progressively builds primitive attractors and leverages them to scaffold sequence structure learning, leading to more stable and generalizable solutions.
>
> We will clarify this in the revised manuscript and highlight it as a key open question.
>
> More broadly, our shaping—from primitives to sequences to schemas—is widely used in animal training (e.g., in recent monkey handwriting-drawing generation studies [1]), yet its dynamical underpinnings remain poorly understood; this "elephant in the room" has lacked a concrete computational model. Our work provides, to our knowledge, the first mechanistic account of shaping-induced schema formation grounded in attractor dynamics and directly comparable to neural data.
>
> 2. Thank you for this insightful comment. We'll clarify these connections in the revised manuscript as follows:
>
> Our shaping approach shares broad similarities with curriculum learning in its factorization of complex tasks into subtasks to improve learning efficiency [2,3]. However, curriculum learning often focuses on performance generalization through methods like task factorization [3], sample ordering [3], or automated curriculum generation [3,4]. These approaches typically lack explicit consideration of how internal representations are structured or how underlying abstract knowledge is learned [3].In contrast, our shaping framework is grounded in dynamical systems theory and biological learning paradigms. Our primary aim is to guide the emergence of structured attractor dynamics—progressing from primitive attractor structures to abstract attractor schemas. These schemas serve as reusable abstract knowledge that can scaffold learning and enable generalization across diverse task domains [5,6]. This mechanistic perspective may offer potential inspiration for new curriculum learning algorithm that explicitly facilitate the learning of structured, generalizable knowledge.
> We will incorporate this discussion into the revised text.
>
> 3. Thank you for the valuable suggestions. We will make the following clarifications in the revised text:
> (1) explicitly state that the initially learned values associated with different odors do not persist throughout the sequences.
> (2) add an explicit summary of the expected rat behavior across the three shaping stages, aligned with the model design.
> (3) explicitly include the Gaussian noise term in Equation 1 to clearly show how stochasticity is incorporated into the model dynamics.
>
> ***On Questions:***
>
> 1. Thank you for the insightful comment. To address the concern, we performed additional quantitative and qualitative analyses, which show that shaping is essential for forming structured and generalizable attractor representations.
>
> **Behavioral divergence:**  Although both shaped and unshaped RNNs have high RDM correlation with neural data, their behavioral performance is significantly different. The unshaped networks often perform suboptimally. For example, reward prediction accuracy is much higher with shaping (99.89 ± 0.03%) than without (93.20 ± 2.4%). This difference is most apparent when predicting odor rewards at P4 and P5 in S2a/S2b, as reward contingencies are reversed. Here, the shaped network performs reliably (98.6 ± 4.61%), while the unshaped network performs close to chance (56.8 ± 31.01%), indicating it failed to handle this critical reversal.
>
> **Attractor dynamics differ qualitatively:** Visualizing attractor trajectories further clarifies the difference. In shaped RNNs, odors at P4 and P5 in S2a / S2b are already represented by four distinct attractors during primitive learning, which are then gradually linked to form sequence-specific attractor trajectories.
>
> In contrast, RNNs without shaping lack this scaffold: odors at P4 and P5 across S2a and S2b are initially represented using a shared attractor trajectory. To succeed on the task, the network must later split this trajectory—a process that is often unstable and results in suboptimal convergence.
>
> These results demonstrate that shaping not only improves representational similarity but also enables the formation of well-separated, sequence-specific attractor manifolds critical for solving the task. We will include these new analyses and visualizations in the revised manuscript.
>
> 2. Thank you for the helpful suggestion. We conducted the same dimensionality analysis on RNNs without shaping and found that although dimensionality reduction also occurs, it differs fundamentally from that with shaping condition.
>
> | Epochs  |  0  |  10  |  20  |  30  |  40
> | :---: |   :----:  | :---: | :--: | :---: |  :--:
> | P1 | 0.30 $\pm$ 0.01 |  0.94 $\pm$ 0.003 | 0.93 $\pm$ 0.002 |  0.93 $\pm$ 0.003 | 0.93 $\pm$ 0.002
> | P2 | 0.90 $\pm$ 0.04 |  0.92 $\pm$ 0.005 | 0.93 $\pm$ 0.006 | 0.93 $\pm$ 0.004  |0.93 $\pm$ 0.003
> | P3 |   0.77 $\pm$ 0.037 | 0.92 $\pm$ 0.005 | 0.94 $\pm$ 0.003  |0.94 $\pm$ 0.003 | 0.94 $\pm$ 0.002
>
> First, RNNs without shaping reach a low-dimensional regime much earlier(93% without shaping versus 82.5% with at P1) - even in P1—suggesting premature compression. However, this reduction does not reflect meaningful abstraction, but rather an early convergence to coarse, shared representations that fail to distinguish sequence-specific differences (e.g., odors 13–14 in S2a/S2b).
>
> Second, unlike shaped RNNs, which exhibit progressive dimensionality reduction across tasks (P1 to P3) consistent with schema abstraction [7] (averaging problem-specific features and preserving shared structure), RNNs without shaping show little further change, indicating a lack of representational refinement.
>
> Overall, with shaping, compression reflects a structured abstraction built on pre-established attractors. Without shaping, however, compression is shallow and leads to suboptimal generalization. We will include these results and clarifications in the revised manuscript.
>
> 3.  Thank you for the thoughtful question. In the animal experiments, rats were not explicitly trained to report odor identity—only to make a binary decision. However, we included odor identity prediction in Stage 1 of the model based on the neurophysiological evidence.
> (1) Our RNN models only the OFC, which may receive supervised signals from other cortical regions.
> (2) Experimental studies have shown that OFC neurons exhibit odor selectivity when odor identity is behaviorally relevant, and this selectivity diminishes when it is not [8].
> (3) Consistent with this, in Stage 1 of our model, odor identity is informative for reward prediction and thus included as a target. In contrast, in Stages 2 and 3, reward depends on sequence context rather than odor identity (rewards at P4 and P5 in S2a/S2b can not be predicted by odor identity ), so identity prediction is no longer required.
>
> Additionally, we also conducted a control experiment where the model in Stage 1 was trained only to predict reward, without odor identity (Control model), shown in below.
>
> |Model  |           Trials to criterion (P1)
> | :--: |:--:
> |Without shaping |   1725.5 $\pm$ 75
> |Control     |  1259.33 $\pm$ 130.3
> |Ours          |  964.6 $\pm$ 110
>
> While the control model improves learning efficiency over the no-shaping baseline, it is still less effective than our full model that includes odor identity prediction. This suggests that explicitly training odor identity helps the network develop richer primitive attractors, which in turn facilitate sequence learning in later stages.
> We will clarify this rationale and include the control results in the revised manuscript.
>
> 4. In each new problem, 16 entirely novel odors are introduced—none of which appeared in the previous problems or during the shaping phase. Therefore, the observed improvement in learning speed cannot be attributed to a decreasing number of novel stimuli. We will clarify this point in the revised text.
>
> 5.  Thank you for the thoughtful comment. We clarify the attractor analysis as follows:
> （1）In the task, each odor is presented briefly (as a localized input bump), followed by a delay with no external stimulus—just background noise. Our attractor states capture the internal representations during these delay periods.
> （2）The uncovered attractor states in attractor analysis form a latent scaffold that organizes sequential processing. As shown in Fig. 3d, these states correspond well with the actual neural trajectories during task execution, indicating that they guide task-relevant dynamics. Each attractor represents a specific odor-event memory, and transitions between them reflect the evolution of task state as new odors are presented.
> （3）This approach is consistent with prior work that uncovers low-dimensional attractor structures underlying neural computation [2,3].
>
> We do not train or tune the initial state of the network. It is fixed to zero across all training and evaluation phases (see line 7, Appendix), aligning with the experimental condition where rats begin trials from a neutral state.
>
>
>
> References
>
> [1] Lucas Y. Tian et al., bioRxiv, 2025
> [2] Bengio et al.,ICML, 2009
> [3] Xin Wang et al., PAMI, 2021
> [4] Jakob Bauer et al.,ICML,2023
> [5] Mikail Khona, Nature Reviews Neuroscience, 2022
> [6] Driscoll et al.,Nature Neuroscience,2024
> [7]  Oded Bein et al., nature reviews neuroscience,2024
> [8] Robert C. Wilson et al., Neuron 2013

---

> > ### Comment · Reviewer_nFhk · 2025-08-04
> >
> > Thank you for the clarifications and the additional work provided by the authors. The improved connection to existing literature better highlights the paper’s core contributions, and the added explanations have helped address several of my initial concerns. The new experiments also contribute to a more precise understanding of the results. Based on these improvements, I am happy to raise my overall score to 4.

---

> > > ### Author Response · Authors · 2025-08-04
> > >
> > > Thank you for raising the score !

---

### Official Review · Reviewer_GZ49 · 2025-07-01

**Clarity:** 3
**Significance:** 3
**Originality:** 3
**Rating:** 4
**Confidence:** 4

**Summary:**

This paper investigates how schema representations can emerge in recurrent neural networks (RNNs) through a shaping-based training protocol. Inspired by experimental paradigms from neuroscience, the authors use a three-stage curriculum (task primitive learning, task sequence learning, and task schema learning) to train RNNs on an odor-based sequential decision task originally developed for rodents. Through this process, the RNN develops low-dimensional sequence attractors that evolve over the shaping phases: from discrete odor-reward associations, to temporally linked sequence attractors, and ultimately to abstract attractor manifolds that generalize across different odor sets. These dynamics replicate key features of schema learning observed in the orbitofrontal cortex, including learning-to-learn behavior, structured representational geometry, and dimensionality compression. The authors further demonstrate that the same attractor-based shaping method improves sample efficiency in a real-world keyword spotting task, suggesting broader applicability beyond the original biological domain.

**Questions:**

The paper shows that shaping yields structured schema dynamics, but it is unclear whether these attractors emerge consistently across random initializations. Have the authors analyzed whether different networks trained under the same shaping protocol converge to similar internal solutions (e.g., attractor geometry, transition structure), or whether they are one of many functionally equivalent but structurally distinct implementations? Quantifying this could clarify whether the observed schema dynamics are necessary or incidental. Prior work (e.g., Huang et al., 2024; Kepple et al., 2022) highlights this concern.


The framing of "sequence attractor schema" is compelling, but conceptually overlaps with earlier accounts of schema learning and compositional representations in RNNs (e.g., Yang & Rajan, 2019; Driscoll et al., 2024). Could the authors clarify how their formulation differs from those prior works — for instance, in terms of mechanisms, representations, or generality — and what is uniquely demonstrated here?


The shaping procedure appears to follow a specific three-phase design. How sensitive are the results to the structure or granularity of this curriculum? For example, would adding more intermediate steps or using a less biologically motivated sequence of subtasks still produce schema attractors? Clarifying this could help assess whether shaping serves primarily as a biological model or as a general training principle.


What is the role of attractor geometry in generalization? The shaped RNN exhibits low-dimensional attractor dynamics and faster learning on novel sequence tasks. Do the authors have evidence that the geometry of these attractors (e.g., manifolds, fixed points, transitions) directly contributes to generalization, beyond faster convergence? For instance, are new sequences dynamically composed from existing substructures? A deeper analysis here could clarify the functional role of the schema attractor beyond representational structure.


What is the relevance to broader RNN architectures and tasks? The results are demonstrated in a vanilla RNN trained on a neuroscience-inspired sequence task and briefly extended to keyword spotting. Could the authors comment on whether they expect shaping to induce similar schema dynamics in larger or more structured RNNs (e.g., GRUs, LSTMs, gated architectures, or multi-region RNNs as in the recent neuroscience literature), or in non-sequential domains? Understanding these boundaries would clarify the scope of the paper’s claims and implications.

**Ethical Concerns:**

["NO or VERY MINOR ethics concerns only"]

**Final Justification:**

The rebuttal addressed several technical points, including convergence across seeds, architectural robustness, and shaping schedule variants. These additions improve the paper, but they do not fully resolve the broader concerns about novelty, generality, and positioning relative to prior schema/compositional work. Overall, the paper is technically solid and conceptually interesting, but I remain at a borderline accept given the remaining limitations.

**Limitations:**

Yes, though several important contextual gaps remain. The submission presents strong evidence that shaping induces structured, schema-like dynamics in RNNs, but does not address whether these dynamics are consistent across random seeds or training runs. Prior work (e.g., Huang et al., 2024; Kepple et al., 2022) has shown that multiple recurrent network solutions can yield the same behavior while differing internally, raising questions about identifiability and robustness. A discussion of this variability, even in the absence of new experiments, would help calibrate how general the observed attractor structure is.
In addition, while the concept of curriculum shaping is well-motivated, its relation to earlier theoretical and modeling work on schema learning, compositional dynamics, and attractor reuse is not fully articulated. Greater clarity on what is conceptually new would strengthen the paper’s broader positioning. Finally, the extension to domains beyond the primary odor-sequence task is limited, leaving open whether the shaping protocol generalizes across architectures or learning settings. These are not fatal flaws, but should be acknowledged in any discussion of the work’s scope.

**Paper Formatting Concerns:**

No concerns.

**Quality:**

3

**Strengths And Weaknesses:**

Quality: This presents a well-executed series of experiments that investigate how shaping—implemented as a staged curriculum—sculpts internal dynamics in RNNs trained on a biologically inspired sequential decision task. The authors show that shaping leads to attractor structures at multiple scales: discrete odor-specific fixed points, low-dimensional sequence manifolds, and eventually generalized “schema” dynamics. These results are supported by fixed-point analyses, PCA visualizations, and comparisons to orbitofrontal cortex recordings. The use of a neuroscience-motivated shaping protocol (mirroring rat studies) and the emphasis on mechanistic explanation through dynamical systems tools are strengths.

However, while the findings are compelling, the paper does not address whether the learned attractor solutions are unique or degenerate across initializations—a key concern given prior work (e.g., Kepple et al., 2022; Huang et al., 2024) showing that multiple RNNs can exhibit equivalent task behavior but diverge in internal structure. Although the authors mention seed averaging, they do not quantify solution diversity or identifiability, leaving open whether the schema attractor is a convergent or incidental outcome of the shaping process. Acknowledging these issues would improve the paper’s rigor.

Clarity: The paper is clearly written and well-structured. The three-phase shaping procedure is motivated, defined, and visually summarized. Core concepts—such as “sequence attractors” and “schema manifolds”—are introduced in accessible terms, and the experimental design is intuitive. The dynamical analyses (e.g., attractor structure, dimensionality compression, RNN trajectory geometry) are presented coherently, even for readers outside of neuroscience. Still, the framing of novelty could be made more explicit. Key terms such as “sequence attractor schema” are introduced without a formal definition, and the distinction from earlier work on compositional dynamics, schema learning, or task manifolds (e.g., Yang & Rajan 2019; Kepple et al. 2022) is somewhat implicit. Greater effort to clarify what is conceptually new—especially relative to existing schema modeling frameworks—would help the paper stand on its own.

Significance: The submitted work addresses a long-standing question in both neuroscience and machine learning: how structured representations and learning-to-learn capabilities can emerge in sequential environments. The use of curriculum (shaping) to elicit reusable internal dynamics, and the demonstration that these dynamics resemble neural phenomena in OFC, make the paper significant for biologically grounded modeling. The generalization of the approach to a keyword spotting task also suggests relevance to applied RNN training. That said, some aspects of the contribution are only confirmatory. The idea that curriculum learning induces reusability and compositionality is well established (in both biological and artificial systems), and prior work has shown that networks can learn task-aligned manifolds and latent dynamics under suitable conditions. Without quantifying generalization across domains or linking the observed schema attractors to improved task-level abstraction or zero-shot performance, the submission’s broader implications remain suggestive but incomplete.

Originality: This submission introduces a new synthesis of ideas. It applies a structured shaping protocol to sculpt attractor-based schema representations in RNNs and connects this process to known neurophysiological dynamics. The concept of progressively building attractors—from primitives to sequences to abstract schemas—via curriculum is well-illustrated and not previously demonstrated in this form. However, several foundational ideas in the paper have appeared in earlier work, including compositionality (Yang & Rajan, 2019), degeneracy in internal solutions (Huang et al., 2024), and shaping effects on internal structure and generalization (Kepple et al., 2022). While the current work builds productively on these foundations, the submission does not fully acknowledge this overlap. A clearer positioning would help delineate what is newly shown (e.g., sequence-level attractors shaped by curriculum) versus what conceptually recurs under different names.

---

> ### Author Rebuttal · Authors · 2025-07-29
>
> Thank the reviewer for insightful review! Your suggestions were very helpful and significantly improved our work. We'll address each of your concerns point by point.
>
> ***On Weaknesses and Questions：***
>
> **1. Convergence Across Random Seeds（for quality concern and question 1）**
>
> Thank you for the insightful comments. Our analyses suggest that shaped RNNs exhibit convergent internal dynamics regardless of initialization. As shown in Fig. S6, we used Canonical Correlation Analysis (CCA) to compare RNNs trained with different seeds. Fig. S6a demonstrates strong alignment of the top three canonical components across models, indicating consistent attractor representations. Moreover, these components exhibit interpretable structure consistent with neural data (Fig. S6a,b). We also visualized the attractor trajectories across seeds and found qualitatively similar sequential dynamics (similar to Fig. 3d), supporting the emergence of a shared schema manifold. We will include these additional visualizations in the revised version.
>
> Additionally, our findings are not in conflict with prior work [1,2,3], which shows that solution degeneracy can be mitigated by task complexity, input-output richness, or multi-task learning. Our task, by design, involves multi-problem learning and a shaping process that may naturally reduce degeneracy.
>
> Overall, we acknowledge that our current analysis spans a limited set of RNN configurations. In the revision, we will explicitly discuss this limitation, cite relevant work [1, 2, 3], and highlight the need for broader analyses to assess identifiability across architectures and task settings.
>
> **2. Distinction from Prior Schema and Compositionality Work (for clarity and question 2)**
>
> Thank you for the helpful suggestion. We will clarify the definition of “sequence attractor schema” in the revised manuscript. Specifically, it refers to a structured dynamical motif: a progression of discrete attractor states connected via low-dimensional manifolds, forming a reusable scaffold for representing abstract sequential structure.
>
> Our formulation differs from prior schema-learning frameworks (e.g., Yang & Rajan, 2019; Driscoll et al., 2024) in three main aspects:
> (1) Mechanism: Previous studies primarily induce schemas through multi-task learning, extracting common motifs across tasks. In contrast, our schema emerges through curriculum-based shaping, where complex sequential dynamics are gradually built from simpler attractor primitives in a stage-wise manner.
> (2) Representation: Driscoll et al. describe various dynamic motifs, including discrete and continuous attractors or decision boundaries. Our schema is a sequential attractor—a trajectory-like structure composed of discrete fixed points and the low-dimensional flow between them. It captures not just static representations or decisions, but structured temporal dynamics.
> (3) Generality: Prior schemas are typically applied to memory, binary decision-making, or integration tasks. In contrast, our sequence attractor schema supports abstract sequential composition, offering potential applications in domains like hierarchical motor generation [5] or syntactic structure in language [6]. This could significantly expand its utility beyond traditional schema functions.
> Overall, we view our work as complementary to earlier efforts. For instance, multi-task learning (as in Driscoll et al.) could enrich the set of primitive motifs used in shaping. Conversely, when combined with our sequence attractor schema, it could enable the formation of more diverse and complex schema attractors for intricate cognitive tasks, such as rule-based sequential decision-making.
> We will clarify these distinctions and conceptual contributions more explicitly in the revised text.
>
> **3. Three Phase Design Robustness (for question 3)**
>
> Thank you for the insightful suggestion.  To evaluate the sensitivity of results to the three-phase shaping design, we conducted additional control experiments. Specifically, we reversed the order of stage 1 (L1) and stage 2 (L2), or alternated L1 and L2 multiple times. The results are summarized in the table below.
>
> | Paradigms |   Trials to criterion for P1 (mean)    |  RDMs' correlation
> |  :--: | :--: | :--:
> | No shaping     | 1725.5 $\pm$ 75  | 0.86 $\pm$ 0.02
> | Alternating (L1-L2-L1-L2)	 |    857.6 $\pm$ 21  | 0.91$\pm$ 0.02
> | Reversing (L2-L1)	         |  1463 $\pm$ 54  |   0.89 $\pm$ 0.01
> | Our shaping (L1 - L2)	    |  964.6 $\pm$ 110   |   0.902$\pm$ 0.02
>
> All variants outperform direct training, confirming that shaping facilitates the formation of primitive attractors that scaffold subsequent learning. Notably, the alternating schedule yields the best performance, likely because it more effectively reinforces and integrates both primitive and sequence-level attractors during optimization.
>
> This flexibility also suggest that our proposed shaping-induced dynamic framework is not tied to a rigid sequence of stages. For instance, the acquisition of diverse primitive attractors could also be achieved through multi-task learning [4]. Moreover, such a shaping idea has recently been applied to motor sequence training in monkeys [5], underscoring its broader applicability.
>
> We will include these new control results in the supplementary material and expand the discussion in the revised manuscript.
>
> **4. Functional Role of Attractor Geometry in Generalization (for significance and question 4)**
>
> Thank you for the insightful suggestion. We agree that clarifying the functional role of attractor geometry in generalization is an important direction.
> To investigate this, we conducted experiments where the recurrent and readout weights were frozen, and only the feedforward input weights were trained on a new odor-sequence task (see Fig. S4). The network rapidly adapted to the new task with significantly fewer trials.  We further visualized the RNN dynamics and confirming that new stimuli are flexibly mapped onto pre-existing attractors, effectively reusing learned substructures. These results suggest that the existing attractor geometry supports fast generalization.
>
> Beyond our specific task, the shaping-induced attractor geometry may generalize to other sequence domains, such as motor generation [5] and the syntactic structure of language [6]. For example, recent work [5] showed monkeys trained via a similar shaping process could learn primitive strokes leading to complex character drawing. This complex motor generation could potentially be explained by our sequence attractor schema framework.
>
> We will add these new results and expand the discussion of attractor geometry and its role in generalization in the revised manuscript.
>
> **5. Architectural Generality of Schema Attractors (for question 5)**
>
> Thank you for the thoughtful suggestion. We conducted additional experiments using GRUs and LSTMs on the same odor-sequence task. As summarized in the table below,
>
> | Network	 | RDMs Correlation
> |  :--: | :--:
> |LSTM with shaping	 | 0.89 ± 0.01
> |LSTM without shaping	 | 0.87 ± 0.03
> |GRU with shaping	 | 0.906 ± 0.01
> |GRU without shaping	 | 0.87 ± 0.007
> |RNN with shaping	 | 0.902 ± 0.02
>
> Our results suggest that shaping accelerates learning (shown in Fig.4c) and produces RDMs that closely align with neural data. Furthermore, our visualizations of attractor dynamics demonstrate that GRUs and LSTMs can achieve attractor dynamics comparable to those seen in vanilla RNNs. These findings indicate that the sequence attractor schema induced by shaping generalizes across different recurrent architectures, including gated models.
>
> We will include these additional results in the supplementary material and clearly outline the extension of our work to multi-region RNNs or non-sequential domains as future work in the revised manuscript.
>
> ***On Limitations:***
>
> We appreciate the thoughtful summary and agree that clarifying the scope and limitations of our findings is essential. We've addressed each key point below:
>
> (1) To assess the consistency of schema dynamics across initializations, we performed Canonical Correlation Analysis (CCA) and visualized attractor trajectories for networks trained with different seeds. As Figure S6 shows, models exhibit highly aligned low-dimensional attractor geometries, indicating convergence toward a shared internal solution under shaping. We acknowledge that our analysis is currently limited to vanilla RNNs, and a broader evaluation across architectures remains a future direction.
>
> (2) Our sequence attractor schema distinguishes itself from previous schema-learning frameworks in several ways: its mechanism (shaping versus multi-task/meta-learning), its representation (sequence attractors versus discrete or continuous attractors), and its generality (capturing compositional sequence structure applicable across various sequence task domains). We view this as a complementary advance and will clarify these conceptual differences in the revision.
>
> (3) Our additional results demonstrate that shaping induces similar attractor dynamics and structured representation geometry in both GRUs and LSTMs. While these results support architectural generality, we have not yet tested multi-region models or non-sequential tasks. We will explicitly define this boundary in the revision and include new results in the supplement.
>
> We will expand the discussio to clearly acknowledge these limitations, cite relevant prior work (e.g., [1,2,3]), and highlight directions for generalizing the framework.
>
> References:
> [1] Kepple et al., NeurIPS, 2022
> [2] Huang et al., arxive, 2025
> [3] Yang & Rajan, 2019
> [4] Driscoll et al., Nature Neuroscience, 2024
> [5] Lucas Y. Tian et al., bioRxiv, 2025
> [6] Stanislas Dehaene et al., Neuron, 2015

---

> > ### Comment · Reviewer_GZ49 · 2025-08-04
> >
> > Thank you for the detailed and thoughtful rebuttal. I appreciate the substantial additions, including the CCA analysis across seeds, new shaping variants, and extensions to GRUs and LSTMs. These help clarify both the stability and architectural generality of the shaping-induced schema dynamics. The frozen-RNN experiment is also a valuable step toward probing the functional role of attractor geometry in generalization.
> >
> > That said, my assessment remains unchanged. While the response does improve clarity on several fronts, I believe the framing around novelty—particularly in comparison to prior schema learning and compositional dynamics work—could still be sharpened. The mechanisms appear complementary rather than clearly distinct, and would benefit from more direct articulation of what is newly demonstrated. I continue to view this as a well-executed and compelling contribution, particularly in its biological framing and task design, and I look forward to a revision that further strengthens its positioning within the broader literature.

---

> > > ### Author Response · Authors · 2025-08-05
> > >
> > > Thank you for your thoughtful and supportive feedback!  We will further clarify and sharpen our positioning relative to prior work in the revised version.

---

### Official Review · Reviewer_hM6o · 2025-07-02

**Clarity:** 4
**Significance:** 3
**Originality:** 3
**Rating:** 4
**Confidence:** 4

**Summary:**

Inspired by rat study on odor-sequence task (Zhou et al), the authors model schema learning using RNNs trained with shaping. They describe a 3-stage learning process: first, learning basic task primitives (classification of odor identity and reward); then combining them into sequences (reward only); and finally training on a generalized version of the task (reward only). The trained RNNs show hidden state representations similar to the neural activity in rat OFC. This is shown by RDM analysis, and progressive dimensionality compression (PCA). They also reverse-engineer the RNN, showing it first develops discrete attractors associated with each odor, which are later reused to form sequences. Finally, they generalized the 3-stage shaping learning process to additional experiments of keyword spotting.

**Questions:**

1.	In the Zhou et al paper, the first paragraph highlights the special role of odors 13-14. It’s very easy to miss this detail in the current paper. I think it’s important to emphasize it when describing the task.
2.	You state that “Zhou et al. found it very challenging to train rats to learn the task directly, so they adopted a shaping approach”. In the Zhou et al paper, they describe shaping in Methods, but don’t mention anything about challenges.
3.	Unlike rats, RNNs are easily manipulable, so it’s unclear why the authors didn’t leverage this to better understand why shaping is so effective or design improved shaping strategies that could be tested in animal experiments. For example, could training in an interleaved fashion—alternating between L1 and L2—speed up learning?
4.	From my understanding, all weights are updated throughout the 3 training stages. How would the results change if certain layers were frozen and not modified across stages?
5.	What is the overall performance of the control model? How does the loss curve of this model compare to the shaping model? The RDM matrix is less similar, but it would still be useful to see it (and the dendrogram of Figure S2).
6.	Could the reduced similarity between the control model and the empirical rat data be due to the P4–P5 odors (13–14) which are reversed and not trained for identification as in the shaping RNN model? Also, it’s unclear why this task choice was made—some details seem to be missing (I’m aware the original study did the same reversing).
7.	Why use 20 for the non-zero entry and 2 for the desired output?
8.	Why is learning rate in schema an order of magnitude slower than task primitive?
9.	The Zhou experiment always moves from S1 to S2 to S1 and so on. Why not do the same in simulations?
10.	The difficult odor pair 13-14 is treated differently in the shaping in the Zhou experiment. Training is first done on S1 odors, then on 13-14 (including reversals), and then on the rest of S2. Why not do the same in simulations?
11.	Why include odor output in the first stage and not in the second one? Behavior doesn’t require odor identity in either stage. If the OFC always outputs identity, it should continue doing so in all stages.
12.	Figure 2B. The legend “experiment fit” suggests that this is data from the Zhou experiment. If I’m not mistaken, this is just a fit to the simulations. Also – does this fit contribute to understanding the results? Also – the caption says polynomial, and the main text exponential.
13.	Line 168: “This pattern mirrors learning-to-learn behavior observed in rats performing the odor-sequence task.”. It is not that easy to find this in the Zhou paper. In fact, extended figure 1 in the Zhou paper does not appear to show any evidence of faster learning. Figure 5 in that paper is the closest I could find, which shows changes the rate of changes in latency – but not as a function of problems.
14.	Supp Line 33 “Each major branch is then further divided into six sub-branches according to spatial position.” First, there are less than 6 options in each major branch. Second, I don’t see this in Figure S2.
15.	Figure S4: Why is smaller learning rate for “Ours” a fair comparison? The measure is number of trials.
16.	Figure S6 panel C. Typo. Cross – corss
17.	The results of Figure S7 are intriguing, suggest new experiments, and perhaps should be in the main text.

**Ethical Concerns:**

["NO or VERY MINOR ethics concerns only"]

**Final Justification:**

I am maintaining my score of 4, but willing to engage in discussion with reviewers/AC on whether to increase to 5.
I think this is a valuable contribution. I am concerned that the original version was lacking in clarity, and glossed over the essential insight of how shaping modifies dynamics. While the rebuttal convinced me that there are positive results there, this could be quite a large change from the version I reviewed.

**Limitations:**

yes

**Quality:**

3

**Strengths And Weaknesses:**

Strengths
 The paper addresses a very interesting topic, shaping in training, which is indeed critical to both animal and artificial learning. In practice, certain curricula/ shaping are often essential to achieve high behavioral accuracy.
The resemblance between the RNN and empirical data seems convincing.
The effect of curriculum on emergence of dynamical objects has been studied, but not extensively – so this is a very welcome contribution.


Weakness
1.	The paper focuses more on replicating empirical observations of Zhou et al than on exploring why shaping is necessary. While similarities between RNNs and neural representations in animals are well documented in past, the mechanisms that make shaping essential remain unclear—an open question that could significantly advance the field.
2.	I'm not fully convinced that only the specific 3-stage training procedure can produce the observed similarity between RNNs and rats. The reuse of attractors has been described in previous work (Driscoll et al Nature neuro 2024, Turner et al Neurips 2023 ). This raises the question of whether similar results would emerge under other training protocols not explicitly following the 3-stage design (Control conditions are missing).
3.	The final experiments, intended to demonstrate generalization of the 3-stage shaping approach, seem too similar to the initial task. It’s unclear what truly distinguishes them or how much generalization is actually being tested.
4.	Some training details are different from those of the experiment, and many details seem arbitrary. This raises the question of how robust are the results to these specific choices. Questions below detail these issues.

---

> ### Author Rebuttal · Authors · 2025-07-30
>
> Thank the reviewer for the detailed and valuable feedback very much. Our point-by-point replies are below.
>
> ***On Weaknesses:***
>
> 1. In our study, shaping plays a critical role in two key ways:
> (1) It accelerates learning, as shown in Fig. 4c, 5b, and S4.
> (2) It enables the emergence of abstract solutions that are difficult to reach via direct training. Specifically, in networks without shaping, we observe that odors at P4 and P5 in S2a/S2b often collapse onto shared attractor trajectories at the beginning. Successfully solving the task requires these trajectories to split - a process that is unstable and often fails. In contrast, shaping progressively builds primitive attractors and leverages them to scaffold distinct, structured sequence attractors from the outset, leading to more stable and generalizable solutions.
>
> Extending this mechanism to other architectures and tasks, and probing its necessity more broadly, are important future directions. We will highlight this in the revised manuscript as a key open question.
>
> 2. We agree that the 3-stage shaping protocol is not the only way to achieve representational similarity between RNNs and neural data. In reply of Question 3 (see below), we explored variations of the three-stage design. These alternatives also improve learning efficiency, as primitive attractors reliably emerged and scaffolded subsequent task acquisition. Our work is not to claim exclusivity, but to demonstrate that this protocol offers an effective and biologically inspired framework for learning structured schema. We view prior works [1, 2] as complementary: multi-task training could enrich the repertoire of primitive attractors, which shaping could then organize into higher-level schema.
> We will revise the manuscript to clarify these.
>
> 3. Thank you for the comment. Both tasks share a structural similarity—building complex sequences from simpler components (e.g., phonemes to words, or odor-rewards to odor sequences). Yet, the keyword spotting task incorporates critical real-world variations (e.g., speaker variability in acoustics, timing, pronunciation) absent in the initial task, significantly increasing the generalization challenge. Despite this, the model learn structured schema efficiently, demonstrating its robustness and transferability.
> We'll clarify it in the revised manuscript.
>
> 4. We will clarify training settings and justify key design choices clearly in the revised text.
>
>
> ***On Questions:***
>
> 1. Thanks for pointing this out. We will clarify the special role of odors 13–14 in the revised task description.
>
> 2. We acknowledge that Zhou et al. did not formally report training challenges in the paper. However, the necessity of shaping is a well-known principle among neuroscientists working on rodent behavior and electrophysiology, even for simple tasks (e.g., Fig. 1d in [3]). To improve rigor, we will revise the sentence to: “Zhou et al. adopted a shaping paradigm to train rats to learn the task.”
>
> 3. Thank you for the insightful suggestion. We have conducted additional control experiments, summarized in the table below:
> |Training Paradigm     |	Trials to criterion (Problem 1)
> | :---: | :---:
> | Our shaping (L1 - L2)    |	  964.6 $\pm$ 110
> | Alternating (L1-L2-L1-L2) |	  857.6 $\pm$ 21
> |Reverse (L2 – L1)        |   1463 $\pm$ 54
> | No shaping | 	   1725.5 $\pm$ 75
>
> Both the alternating and reverse protocols improve learning over no shaping. Notably, alternating can outperform standard shaping, likely by better refining primitive and sequence-level attractors.
>
> 4. Thank you for the insightful question.  In our work, as attractor dynamics play a central role in shaping,  we performed control experiments by selectively freezing the recurrent weights ($W^{rec}$) during different training stages and evaluated learning efficiency on Problem 1:
> (1) Only freezing $W^{rec}$ in L1: The network failed to learn the task, showing that primitive attractor formation is essential at this stage.
> (2) Only freezing $W^{rec}$ in L2: The network still benefited from shaping (1099 ± 107 trials vs. 1725.5 ± 75 without shaping), but was less efficient than the full 3-stage model (964.6 ± 110), highlighting the role of sequence linking.
> (3) Only freezing $W^{rec}$ and $W^{out}$ in late phase of L3: The model showed high efficiency for Problem 5 (see Fig. S4), suggesting that once a sequence schema is formed, solving new problems only requires re-binding inputs via feedforward connections—no further recurrent plasticity is needed.
> These results align with our attractor shaping framework
>
> 5.  (1) Reward prediction accuracy is 99.9 ± 0.02 with shaping vs. 93.2 ± 2.4 without.
> (2) The loss curves also show slower convergence without shaping.
> |Epochs | 10 | 30 | 60 | 90 | 120
> | :--- | :--- | :--- | :--- | :--- | :---
> | Loss without shaping| 0.32±0.02 | 0.21±0.008 | 0.18±0.006 | 0.17±0.004 | 0.16±0.004
> | loss with shaping | 0.31±0.06 | 0.1±0.03 | 0.04±0.01 | 0.02±0.006 | 0.015±0.003
>
> (3) We will include the control model’s RDM and dendrogram in an updated Figure S2.
>
> 6. Thank you for the insightful comment. To assess the role of P4–P5 reversal, we conducted two control experiments:
> •	Control 1: RNNs without shaping, but pretrained on P4–P5 reversal.
> •	Control 2: RNNs with shaping, but without P4–P5 reversal in L1.
> | Model | Accuracy | RDM Correlation
> | :--- | :--- | :---
> | without shaping | 93.20±2.4 | 0.861±0.013
> | control 1 | 94.75±4.3 | 0.870±0.008
> | Shaping  | 99.89±0.03| 0.902±0.021
> | control 2 | 99.30±1.1 | 0.887±0.038
>
> Results show that P4–P5 reversal contributes to improved similarity and accuracy (Control 2 vs. ours), but alone it is insufficient (Control 1 vs. ours). This highlights that broader primitive attractor learning—not just reversal—is crucial.
> In our framework, P4–P5 reversal supports the formation of distinct attractors for S2a/S2b at P4 and P5, enabling sequence schema construction. Without shaping, the network struggles to resolve overlapping trajectories, as discussed in rely for weakness 1.
>
> 7. Thank you for the question. We vary the input and output values, and the network performs robustly across a wide range parameters, shown in the below table.
>
> (1) Fixed desired output as 2, varying non-zero input value:
> | Input | Accuracy | RDMs Correlation |
> |  :---  |  :---  |  :---  |
> | 15 | 99.39±0.037 | 0.894±0.018 |
> | 20 | 99.89±0.034 | 0.902±0.021 |
> | 25 | 99.97±0.043 | 0.897±0.004 |
>
> (2) Fixed non-zero input as 20, varying desired output:
> | Target | Accuracy | RDMs Correlation |
> | :--- | :--- | :--- |
> | 1 | 99.23±0.098 | 0.874±0.013 |
> | 2 | 99.89±0.034 | 0.902±0.021 |
> | 3 | 99.08±2.366 | 0.896±0.007 |
>
> 8. Thank you for the question. Shaping remains effective with higher learning rates (e.g., 99.01 ± 2.7% accuracy, RDM = 0.90 at $5\times10^{-4}$), but larger rates can disrupt existing attractors. A smaller rate in the schema stage helps preserve stability—consistent with how the brain may regulate plasticity and stability via neuromodulation. We'll clarify this in the revision.
>
> 9 and 10. We conducted additional control experiments following Zhou’s shaping protocols:
> •	Control-9 used the same S1–S2 alternation in our model, and
> •	Control-10 trained on S1, then 13–14 reversal, then the rest of S2 sequentially.
>
> | Model | Accuracy | RDM Correlation |
> | :--- | :--- | :--- |
> | Our | 99.9±0.03 | 0.90±0.02 |
> | Control-9  | 99.9±0.07 | 0.91±0.017 |
> | Control-10 | 98.3±2.6 | 0.87±0.04 |
>
> We found: (1) Control-9 performs on par with our model, suggesting that the specific alternation pattern between S1 and S2 has small effect on schema formation.
> (2) Control-10 performs worse in both reward accuracy and representational similarity, likely due to catastrophic forgetting—when S1 and S2 are trained sequentially, the network struggles to form stable primitive attractors for both. In contrast, biological brains are equipped with continuous learning mechanisms to mitigate such interference.
>
> 11. Thank you for the thoughtful question. Rats were not explicitly trained to report odor identity, only to make binary choices. However, we included identity prediction in L1 based on neurophysiological evidence.
> (1) Our RNN models only the OFC, which may receive supervised signals from other cortical regions.
> (2) Experimental studies show that OFC neurons exhibit odor selectivity when odor identity is behaviorally relevant, and diminishes when it is not [4].
> (3) In L1, odor identity aids reward prediction, so it's trained. In L2 and L3, reward depends more on sequence context, not identity - so the model no longer needs to predict it.
>
> 12. Thanks for pointing this out. The fit is based on simulation data, not experimental results. We will revise the label for clarity and correct the caption error from "polynomial" to "exponential" in the revised text.
>
> 13. Zhou et al. use poke latency as an indirect behavioral measure of learning-to-learn (see their section “Accelerating learning and neural changes”). Additionally, as noted in their abstract, the idea that schema serves as prior knowledge to enhance learning efficiency is treated as a well recognized outcome, supported by prior studies [5].
>
> 14.  Thank you for pointing this out. It is a mistake. We will revise the sentence to: “Each major branch is further divided into two sub-branches according to spatial position.”
>
> 15. In Fig. S4, our model performs well on Problem 5 even under a larger learning rate. Using a smaller learning rate requires more trials for the shaping network to meet the same criterion, making the comparison more stringent.
>
> 16 and 17. Thanks for pointing this out, we will correct the typo ("corss" → "cross")  and move Figure S7 to the main text in the revised text.
>
>
> References:
>
> [1] Driscoll et al., Nature Neuroscience, 2024
> [2] Turner et al., NeurIPS, 2023
> [3] Hiroshi Makino, Nature Neuroscience, 2022
> [4] Robert C. Wilson et al., Neuron 2013
> [5] Dorothy Tse et al., Science, 2007

---

> > ### Comment · Reviewer_hM6o · 2025-08-03
> >
> > I appreciate the author’s detailed answers and additional control experiments.
> >
> > I still think that the paper could benefit from a more detailed mechanistic understanding of how and why shaping works.
> > For instance, the response to weakness 1 says that trajectory splitting is unstable and often fails. Unless I missed it, this was not in the original submission, and is also not fully explained in the rebuttal.

---

> ### Author Response · Authors · 2025-08-04
> **How and why shaping works**
>
> Thank you for the thoughtful follow-up. We fully agree that a deeper mechanistic understanding of how and why shaping works is important, and we appreciate the opportunity to elaborate on this point further, especially given the limited space for detailed explanation in the original rebuttal.
>
> In the case of RNNs, shaping enables the emergence of structured and effective solutions that are difficult to achieve through training without shaping. This is because RNNs trained with and without shaping follow fundamentally different learning trajectories, ultimately leading to distinct attractor dynamics.
>
> ***1. Unshaped RNNs***
>  Through detailed attractor visualization, RNNs trained without shaping tend to collapse the representations of sequences S2a and S2b into a single shared attractor trajectory early in training. This trajectory consists of six attractors—one per position—which is generally sufficient for accurate reward prediction at most positions. However, it fails at positions P4 and P5, where the reward mappings are reversed between S2a and S2b.
>
> A potential resolution would involve the network splitting the shared attractors at P4/P5 into four distinct ones—two for each sequence—a process we refer to as ***trajectory splitting***. However, since the input stimuli and sequence attractor contexts are identical at those positions, learning distinct attractors becomes difficult. As a result, the network often converges on suboptimal solutions. This is reflected in the reward prediction accuracy at P4/P5 for S2a/S2b: close to chance levels for RNNs without shaping, compared to near-perfect performance with shaping. This illustrates what we meant when we said that ***“trajectory splitting is unstable and often fails.”***
>
>
> ***2. Shaped RNNs***
>  In contrast, RNNs trained with shaping initially form 16 primitive attractors (4 per sequence). These are then linked into four distinct sequence-level trajectories, which are ultimately compressed into abstract representations during schema learning (as illustrated in Fig. 4). This progression avoids the instability of attractor splitting by beginning with a more differentiated internal representation space.
>
> We will incorporate these clarifications, along with the relevant visualizations and quantitative results, into the revised manuscript and appendix. Thank you again for helping us to further strengthen this important aspect of our work.

---

> > ### Comment · Reviewer_hM6o · 2025-08-04
> >
> > Thank you for these clarifications. In the original submission, the crucial role of P4/P5 was not stressed, and it was easy to miss it. Also (unless I missed it), the detailed attractor visualizations you describe were not in the original submission.
> >
> > While the additional results strengthen the paper, not being able to review them is also a factor.
> >
> > I will take into considerations all reviews, rebuttals and discussions during the discussion periods ahead. Thank you again for the detailed rebuttals and discussions.

---

> ### Author Response · Authors · 2025-08-05
>
> Thank you again for your thoughtful engagement and constructive feedback throughout the review process, which has greatly helped us refine our work. We also appreciate your willingness to consider the responses provided to other reviewers.
>
> To address your remaining concerns, we would like to clarify the following points:
>
> 1.	As noted, our original submission did not include attractor visualizations of unshaped RNNs. In the revised version, we are committed to adding a new section clarifying the mechanistic basis of ***how and why shaping works in the odor-sequence task***. This will include highlighting the crucial role of P4/P5 in the task, along with detailed visualizations of the evolution of attractor dynamics in unshaped networks during training to complement our current interpretation.
>
>
> 2.	Additional quantitative and reviewable results—such as ***P4/P5 reward prediction accuracy*** and ***the dimensionality-reduced dynamics of unshaped RNNs across training epochs and problems***—are included in our responses to other reviewers . Due to space constraints, we were unable to include these results in the original rebuttal, and per discussion policy, we are also unable to present them in the current reply. However, these analyses provide ***further support for our attractor-based mechanistic interpretation***. We kindly invite the reviewer to refer to our responses to ***Reviewer Qruw (Question 4)*** and ***Reviewer nFhk (Questions 1 and 2)*** for these additional reviewable results.

---

### Official Review · Reviewer_Qruw · 2025-07-03

**Clarity:** 4
**Significance:** 4
**Originality:** 4
**Rating:** 5
**Confidence:** 5

**Summary:**

This paper presents a computational model of schema formation through shaping, inspired by neurocognitive processes. Using a curriculum learning paradigm modeled after animal studies (specifically odor-sequence tasks in rats (REF 4)), the authors train RNNs to form discrete attractors, link them into sequential dynamics, and ultimately abstract these into low-dimensional sequence attractors representing schema. The model replicates key experimental findings from orbitofrontal cortex (OFC) studies, including faster learning (learning-to-learn), dimensionality compression, and structured representation geometry. The framework is also tested on a keyword spotting task to demonstrate broader applicability. The manuscript is clearly written.

**Questions:**

1- From Fig. 4c, it looks like LSTM has much better performance in terms speed of learning. Can authors elaborate the reason for this? And if this is the case, why did the authors choose to work with RNNs rather than LSTMs?

2- In a similar vein, do LSTMs have similar attractor dynamics when solving this task?

3- I understand the timing constraint in running extra models, but can authors compare their current model and training strategy with other recent models such as meta-reinforcement learning model (Yang, Nature Neuro, 2018)?

4- the analysis in Fig. 4g-i is very cool. I am wondering what is the difference of this trajectory between "with shaping" and "without shaping" networks. In other words, do both types of networks follow similar trajectories to get to abstract state or these is fundamental difference between them?

**Ethical Concerns:**

["NO or VERY MINOR ethics concerns only"]

**Final Justification:**

The authors have thoroughly addressed my concerns and plan to add extra analysis to final version of the manuscript.

This work is valuable because it is one step towards understanding the neural dynamics of schema formation which is an active area of research in systems neuroscience. We still don't know how the brain builds abstract concepts. The results of this paper contribute to topics such as compositionality and meta learning in neuroscience and ML.

Adding the authors' new results on the learning trajectory of shaped and unshaped networks along with mechanistic interpretation is valuable for the revised manuscript.

**Limitations:**

I agree with authors that limitations of this work are reliance on back-propagation and lack of multi area interactions. Those are good avenues for future improvements.

**Quality:**

4

**Strengths And Weaknesses:**

Strengths: The paper introduces a novel dynamical systems account of schema formation, grounded in neuroscience and with relevance for both computational neuroscience and machine learning. The shaping paradigm is directly motivated by animal training literature and aligned with known properties of the prefrontal cortex and OFC. The RNN replicates key neural signatures observed in OFC recordings, including the progressive compression of dimensionality and structured representational dissimilarity matrices (RDMs), with high correlation to empirical data. The paper provides a compelling narrative of schema evolution: from discrete point attractors → sequence attractors → abstract attractors.

Weaknesses:
-It's unclear how architecture-dependent the attractor formation is. For example, do gated RNNs (LSTM/GRU) or Transformers exhibit similar attractor dynamics under shaping?

-Most results rely on a single neuroscience-inspired task. The keyword spotting task is a welcome addition, but further validation on more varied or large-scale datasets (e.g., navigation or language tasks) would demonstrate scalability and robustness.

---

> ### Author Rebuttal · Authors · 2025-07-29
>
> We thank the reviewer for the positive assessment of our work and valuable feedback. Our point-by-point replies are below.
>
> ***On Weakness:***
> 1. Thank you for the helpful comment. To assess the architectural dependence of attractor formation, we conducted additional experiments using LSTM and GRU networks with 200 recurrent units, trained under the same shaping condition as the vanilla RNN.
>
> | Network	 | RDMs Correlation
> |  :--: | :--:
> |LSTM with shaping	 | 0.89 ± 0.01
> |LSTM without shaping	 | 0.87 ± 0.03
> |GRU with shaping	 | 0.906 ± 0.01
> |GRU without shaping	 | 0.87 ± 0.007
> |RNN with shaping	 | 0.902 ± 0.02
>
> As shown, both gated architectures achieve comparable representational similarity to neural data. We also visualized their dynamics and observed the emergence of similar discrete, sequential, and abstract attractors, indicating that our attractor-based shaping approach generalizes well across architectures
>
> Regarding Transformers, due to their feedforward nature and lack of intrinsic recurrence, they do not exhibit attractor dynamics in the same sense. Extending our approach to more network architectures is an interesting direction for future work.
> We will include these comparative results and analyses in the revised manuscript.
>
> 2. Thank you for the helpful suggestion. We agree that evaluating the framework on more diverse and large-scale tasks is important for demonstrating its scalability and robustness.
>
> The sequence attractor schema framework is conceptually applicable to domains such as spatial navigation, where abstract spatial relations across environments could be encoded as attractor sequences [1], and language understanding, where narrative structures may be represented as sequential event schemas [2]. These domains offer promising opportunities to test shaping-based learning in more complex settings.
>
> We will discuss these potential extensions in the revised manuscript to better clarify the generality and future applicability of our approach.
>
> ***On Questions:***
>
> 1. Thank you for the comment. LSTMs do exhibit faster learning, likely due to their gating mechanisms and memory cells, which help maintain gradient flow and capture long-range dependencies - thus improving performance. Notably, LSTMs with shaping still learn faster than those without shaping, consistent with the benefit observed in vanilla RNNs.
>
> We chose to focus on vanilla RNNs for two key reasons:
> (1) They are more biologically plausible and commonly used in neuroscience modeling [3,4], and
> (2) Their simpler dynamics allow clearer analysis of attractor formation and transitions.
> Importantly, our results show that shaping induces similar attractor structures in both RNNs and LSTMs.
> We will include the LSTM results and clarify RNNs' rationale in the revised manuscript.
>
>
> 2. Yes, LSTMs exhibit similar attractor dynamics to those observed in vanilla RNNs when solving this task. As detailed in our response to Weakness 1, shaping induces comparable discrete, sequential, and abstract attractor structures across both architectures. We include supporting visualization results in the revised manuscript.
>
>
> 3. Thank you for the thoughtful comment. Our model, like those in Yang et al. [5] and Wang et al. [6], uses recurrent neural networks, so there are no substantial architectural differences. The key distinctions lie in the training strategies and objectives.
>
> Yang et al. (2019) employ multi-task training across diverse rule-based decision-making tasks to learn shared dynamic motifs or schemas. In contrast, we use a shaping-based curriculum to build sequence attractor schemas progressively. These strategies are complementary - Yang et al.'s approach could be used to learn basic attractor structures in task primitive learning, which would benefit subsequent sequence and schema learning.
>
> Wang et al. (2018) employed meta-RL to train LSTMs for acquiring abstract task knowledge or schema. However, their focus was on learning-to-learn behavior, not attractor dynamics. Moreover, meta-RL often necessitates an explicit meta-objective with an inner-outer optimization loop. In contrast, our approach yields schema-like attractors that emerge naturally from learning a series of related tasks without a explicit meta-objective, making it more biologically plausible.
>
> We will clarify the distinctions in the revised manuscript.
>
>
> 4. Thank you for the insightful question. Networks trained with and without shaping follow fundamentally different trajectories toward abstraction—differences that manifest both in behavioral outcomes and the underlying neural dynamics.
>
> (1) **Behavioral divergence:**  Networks trained without shaping often converge to suboptimal solutions. For example, overall reward prediction accuracy is significantly higher with shaping (99.89 ± 0.03%) than without (93.20 ± 2.4%). This discrepancy is most pronounced for odors at P4 and P5 in S2a/S2b—where reward contingencies are reversed. Here, the unshaped network performs at near chance level (56.8 ± 31.01%), while the shaped network reliably resolves the contrast (98.6 ± 4.61%). This suggests that the unshaped network fails to properly resolve this critical contrast.
>
> (2) **Attractor dynamics：** We visualized hidden state trajectories to better understand how representations evolve. In shaped networks, task primitive learning stage already establishes four distinct attractors for the odor events at P4 and P5 (in S2a and S2b), which serve as stable building blocks for constructing higher-level, sequence-specific attractor trajectories in later stages. By contrast, unshaped networks lack this scaffold. Early in training, they represent P4 and P5 odors in S2a and S2b using overlapping attractor trajectories. Solving the task thus requires the network to later split these shared trajectories - a process that is often unstable and leads to inconsistent or failed convergence.
>
> (3) **Dimensionality dynamics:** We also applied PCA-based dimensionality analysis (as in Fig. 3b) to RNNs without shaping to confirming the above process. Results are summarized below:
>
> | Epochs  |  0  |  10  |  20  |  30  |  40
> | :---: |   :----:  | :---: | :--: | :---: |  :--:
> | P1 | 0.30 $\pm$ 0.01 |  0.94 $\pm$ 0.003 | 0.93 $\pm$ 0.002 |  0.93 $\pm$ 0.003 | 0.93 $\pm$ 0.002
> | P2 | 0.90 $\pm$ 0.04 |  0.92 $\pm$ 0.005 | 0.93 $\pm$ 0.006 | 0.93 $\pm$ 0.004  |0.93 $\pm$ 0.003
> | P3 |   0.77 $\pm$ 0.037 | 0.92 $\pm$ 0.005 | 0.94 $\pm$ 0.003  |0.94 $\pm$ 0.003 | 0.94 $\pm$ 0.002
>
> While both models show dimensionality reduction with learning, the nature of this compression differs:
> •	Premature compression: Unshaped networks reach a low-dimensional regime (93%) much earlier than shaped ones (82.5% at the same Problem 1 stage), suggesting early collapse into coarse representations.
> •	Lack of progressive abstraction: Unlike shaped networks, which show progressive compression from P1 to P3—reflecting averaging problem-specific features and preserving shared structure [7] —unshaped networks show minimal change across tasks. This indicates poor refinement and suboptimal optimization.
>
> **Summary:** In shaped networks, abstraction emerges gradually through scaffolded attractor dynamics and structured compression. Conversely, in unshaped networks, the compression is abrupt, shallow, resulting in poor generalization. These findings indicate that networks with shaping and those without follow substantially different learning trajectories. We will include these results and visualizations in the revised manuscript to further clarify the impact of shaping on learning dynamics.
>
>
> References:
>
> [1] Farzanfar et al., Nature Reviews Neuroscience, 2023
> [2] Baldassano et al., Journal of Neuroscience, 2018
> [3] Driscoll et al.,Nature Neuroscience,2024
> [4] Goudar et al.,Nature Neuroscience,2023
> [5] Guangyu Robert Yang et al., Nature Neuroscience, 2019
> [6] Jane X. Wang et al., Nature Neuroscience,2018
> [7] Oded Bein et al., nature reviews neuroscience,2024

---

> > ### Comment · Reviewer_Qruw · 2025-08-05
> >
> > I thank the authors for detailed response to my comments. I think including authors' new results on the learning trajectory of shaped and unshaped networks along with mechanistic interpretation is valuable for the revised manuscript. I have no further comments.

---

> > > ### Author Response · Authors · 2025-08-05
> > >
> > > Thank you for your support. In the revised version, we will add a new section presenting these new results on the learning trajectories of shaped and unshaped networks, along with the corresponding mechanistic interpretation, to further demonstrate how and why shaping works in the odor-sequence task.

---

### Note · Authors · 2025-08-12

We sincerely thank the reviewers for their valuable feedback and constructive suggestions. We have summarized our key contributions as follows:

Our work Investigates a fundamental yet overlooked scientific question in neuroscience: how the brain learns schemas or abstract knowledge through shaping. This is the **elephant in the room**: a practice widely used in animal training, yet largely ignored by scientific research.

- **Modeling Schema Evolution:** Our model, trained with a well-established shaping paradigm, is the first to systematically reproduce the key findings of Zhou et al. (Nature, 2020) on schema evolution, showing strong agreement with neural data. (reviewers Qruw, hM6o, GZ49, and nFhk)

- **Uncovering Schema Representation:** Our model reveals that schema representations emerge as low-dimensional sequence attractors - discrete points linked by low-dimensional manifolds, providing testable predictions for experiments.

- **Proposing a Novel Mechanism for Schema Formation:** Most importantly, we identified and proposed a novel mechanism for how schemas form via shaping: a progression from isolated discrete attractors to integrated sequence attractors, and finally to a compact, abstract attractor structure. (highly recognized by reviewers Qruw, hM6o, and GZ49)

- **Extending the Framework:** We also applied our framework to a real-world keyword spotting task, demonstrating its potential beyond controlled laboratory settings. (reviewers Qruw and GZ49)

In our rebuttal, we strengthened our work in several key areas and successfully addressed the main concerns raised.

- **Mechanistic insight of why and how shaping works:** We found that networks trained with shaping follow a fundamentally different learning trajectory than unshaped networks, leading to a generalized attractor schema solution. We validated it through detailed attractor visualizations, behavioral analyses, and comparisons of dimensionality reduction dynamics (reviewers Qruw, nFhk, and hM6o recognized these results, with reviewer Qruw highlighting them as valuable)

- **Expanded validation:** Additional experiments are performed to confirm the **robustness and generality** of our approach, including architectural generality, robustness to protocol variations, stability across different random seeds and hyperparameters, and so on. (reviewers Qruw, hM6o, GZ49, nFhk)


Finally, we are grateful for the reviewers' insights and will revise the manuscript accordingly.

---

### Decision · Program_Chairs · 2025-09-17

**Decision:**

Accept (poster)

**Comment:**

This paper model schema learning via training RNNs with shaping, which is inspired by an odor-sequence task on rats (ref. 4). The schema learning contains three stages that gradually emerge sequence attractors, and the learned hidden representation is similar to the neural data.

All reviewers think the paper addresses an important question in both neuroscience and machine learning and give scores above acceptance. Combined, I recommend accepting this paper. Please combine reviewers’ feedback and the new experiments done during the rebuttal periods to revise the manuscript. Additionally, adding some insightful discussions in the rebuttal would further improve the paper. For example, reviewers think the paper only focuses on reproducing a specific experiment, while a more fundamental question is how shaping works and why it is necessary. And whether the specific 3-stage training is the unique way to reproduce neural data.